# Glioblastoma Embryonic-like Stem Cells Exhibit Immune-Evasive Phenotype

**DOI:** 10.3390/cancers14092070

**Published:** 2022-04-21

**Authors:** Borja Sesé, Sandra Íñiguez-Muñoz, Miquel Ensenyat-Mendez, Pere Llinàs-Arias, Guillem Ramis, Javier I. J. Orozco, Silvia Fernández de Mattos, Priam Villalonga, Diego M. Marzese

**Affiliations:** 1Institut d’Investigació Sanitària Illes Balears (IdISBa), Cancer Epigenetics Laboratory, Cancer Cell Biology Group, 07120 Palma, Illes Balears, Spain; sandra.iniguez@ssib.es (S.Í.-M.); miguelarash.ensenat@ssib.es (M.E.-M.); pere.llinas@ssib.es (P.L.-A.); 2Institut d’Investigació Sanitària Illes Balears (IdISBa), Institut Universitari d’Investigació en Ciències de la Salut (IUNICS), Cancer Cell Biology Group, Universitat de les Illes Balears, 07122 Palma, Illes Balears, Spain; guillem.ramis@uib.cat (G.R.); silvia.fernandez@uib.es (S.F.d.M.); priam.villalonga@uib.es (P.V.); 3Serveis Científico-Tècnics, Universitat de les Illes Balears, 07122 Palma, Illes Balears, Spain; 4Saint John’s Cancer Institute, Providence Saint John’s Health Center, Santa Monica, CA 90404, USA; javier.orozco@providence.org; 5Departament de Biologia Fonamental i Ciències de la Salut, Universitat de les Illes Balears, 07122 Palma, Illes Balears, Spain

**Keywords:** glioma stem cells, glioblastoma, antigen presentation, transcriptomics, epigenomics

## Abstract

**Simple Summary:**

Most glioblastoma (GBM) patients relapse after an initial response to treatment. These aggressive traits are often associated with the presence of glioma stem cells (GSCs) within the tumor bulk, which are thought to participate in GBM therapy resistance. Given GBM cellular heterogeneity, we hypothesized that GSCs might also display cellular hierarchies associated with different degrees of stemness. Based on single-cell RNAseq data from GBM patients, we identified a subpopulation of GSCs, named core-GSCs (c-GSCs), with a similar profile to embryonic stem cells and downregulation of immune-associated pathways. In addition, we developed an in vitro induced c-GSC (ic-GSC) model resembling their tumor counterpart. The characterization of immune-privileged c-GSCs provides a valuable resource to study immune evasion mechanisms in GBM and to identify potential unexplored targets to improve immunotherapy treatments.

**Abstract:**

Background: Glioma stem cells (GSCs) have self-renewal and tumor-initiating capacities involved in drug resistance and immune evasion mechanisms in glioblastoma (GBM). Methods: Core-GSCs (c-GSCs) were identified by selecting cells co-expressing high levels of embryonic stem cell (ESC) markers from a single-cell RNA-seq patient-derived GBM dataset (*n* = 28). Induced c-GSCs (ic-GSCs) were generated by reprogramming GBM-derived cells (GBM-DCs) using induced pluripotent stem cell (iPSC) technology. The characterization of ic-GSCs and GBM-DCs was conducted by immunostaining, transcriptomic, and DNA methylation (DNAm) analysis. Results: We identified a GSC population (4.22% ± 0.59) exhibiting concurrent high expression of ESC markers and downregulation of immune-associated pathways, named c-GSCs. In vitro ic-GSCs presented high expression of ESC markers and downregulation of antigen presentation HLA proteins. Transcriptomic analysis revealed a strong agreement of enriched biological pathways between tumor c-GSCs and in vitro ic-GSCs (*κ* = 0.71). Integration of our epigenomic profiling with 833 functional ENCODE epigenetic maps identifies increased DNA methylation on HLA genes’ regulatory regions associated with polycomb repressive marks in a stem-like phenotype. Conclusions: This study unravels glioblastoma immune-evasive mechanisms involving a c-GSC population. In addition, it provides a cellular model with paired gene expression, and DNA methylation maps to explore potential therapeutic complements for GBM immunotherapy.

## 1. Introduction

Glioblastoma (GBM) is the most aggressive primary brain tumor with a median overall survival of 15 months [1]. Current treatment involves surgical resection followed by concurrent radiotherapy and temozolomide-based chemotherapy [2]. However, the systematic development of drug resistance traits is associated with poor prognosis, high recurrence, and high mortality rates. Although immunotherapy (IT) has shown promising results against different solid tumors, phase 3 clinical trials have shown limited efficiency for GBM [3]. Tumor heterogeneity and phenotypic plasticity combined with a strong immunosuppressive tumor microenvironment remain a challenge for a successful IT in GBM.

GBM contains a subset of glioma stem cells (GSCs), which possess tumor-initiating capacity and self-renewal potential [4]. Importantly, GSCs have become a potential target for anti-cancer therapies due to their critical role in tumor aggressiveness and survival [5] and their ability to escape from immune recognition [6]. Nevertheless, purification, expansion, and characterization of GSCs remain a challenging endeavor [7], potentially in part due to the selective pressure of the culturing conditions favoring specific types of GSC populations. Given the high degree of cellular diversity observed in GBM, we hypothesized that GSCs might also comprise a heterogeneous population with different states of stemness among themselves. Therefore, we sought to find a highly undifferentiated GSC population within the GBM tumor cell mass.

Embryonic stem cells (ESCs) give rise to a wide range of cell types in the adult organism. Interestingly, several groups have reported that ESC-specific gene regulatory signatures are shared among various cancer types [8,9,10,11]. Moreover, it has been shown that the ESC core transcription factors OCT4, SOX2, and NANOG are present in GBM tumors, and the upregulation of these factors correlates with poor survival [12,13]. Based on single-cell RNA-seq (scRNA-seq) data from GBM tissues [14], we identified a pool of triple-positive GBM cells co-expressing OCT4, SOX2, and NANOG named core GSCs (c-GSCs). Importantly, we observed a significant downregulation of antigen presentation-associated genes in c-GSCs. Using induced pluripotent stem cell (iPSC) technology [15], we established an in vitro cellular model that resembles the tumor c-GSCs. We named these cells induced c-GSCs (ic-GSCs). Additionally, we generated gene expression and DNA methylation (DNAm) maps of ic-GSCs and parental GBM patient-derived cells (GBM-DCs). These data provide hints about the epigenetic and transcriptomic changes involved in immune attenuation mechanisms in ic-GSCs. These findings shed light on the molecular mechanisms underlying immune evasion in GBM.

## 2. Materials and Methods

### 2.1. Single-Cell RNA-Seq Data Analysis

Single-cell RNA-seq (scRNA-seq) data from GBM specimens (*n* = 28) generated by Neftel C et al. was obtained from the Broad Institute Single Cell Portal [14]. All genes with no expression in more than 95% of cells were removed from further analysis. Cells with concomitant high expression of OCT4, SOX2, and NANOG (OSN), defined as the upper quartile expression for each gene, were identified as c-GSCs. Gene Ontology (GO) was used to identify enriched pathways using all differentially expressed genes (DEGs). Radar plots were used to depict HLA-A, -B, and -C (HLA-ABC) levels relative to the OSN profile. R (v.4.0.2) packages plotly (v.4.9.4.1), ggplot2 (v.3.3.2), and fmsb (v.0.7.0) were used for data representation.

### 2.2. Cell Lines and Culture Conditions

GBM-DCs (U3035MG) were obtained from the Human Glioblastoma Cell Culture biobank (Uppsala, Sweden) and cultured under neurobasal media conditions, as previously described [16]. ic-GSCs were cultured on Matrigel-coated dishes (#734-1440, VWR, Radnor, PA, USA) with mTeSR1 media (#85850, STEMCELL Technologies, Vancouver, BC, Canada) as per the manufacturer’s instructions.

### 2.3. Lentiviral Production and ic-GSCs Derivation

FUW-tetO-hOKMS (Addgene plasmid #51543; RRID: Addgene_51543) was a gift from Tarjei Mikkelsen [17] and the FUW-M2rtTA (Addgene plasmid #20342; RRID: Addgene_20342) was a gift from Rudolf Jaenisch [18]. GBM-DCs were reprogrammed into ic-GSCs using the iPSC generation protocol [18]. Briefly, VSVG coated lentiviruses were packaged in 293 T cells cultured in mTeSR1 media. FUW-OKMS and FUW-M2rtTA viral supernatants were mixed at a 1:1 ratio, and 0.3 × 106 GBM-DCs were infected four times for 48h. Transduced GBM-DCs were treated with doxycycline 2 μg/mL (#72742, STEMCELL Technologies) to induce OKMS expression until ic-GSC colonies appeared after four weeks. Single-cell clone selection was conducted following manual serial dilution in 96-well plates.

### 2.4. Immunostaining

Cells were fixed in 4% paraformaldehyde for 20 min. Immunostaining was performed according to standard protocols. Antibodies were diluted in 1X PBS 0.1% BSA. Primary antibodies and concentration: goat anti-SOX2 1:100 (#LSBio (LifeSpan) Cat#LS-C132162-200, RRID: AB_10833714, R&D), goat anti-OCT4 1:200 (R and D Systems Cat# AF1759, RRID: AB_354975), goat anti-NANOG 1:100 (R and D Systems Cat# AF1997, RRID: AB_355097), and mouse anti-HLA-ABC 1:100 (Thermo Fisher Scientific Cat# MA5-11723, RRID: AB_10985125). Secondary antibodies and concentration: donkey anti-goat IgG (Thermo Fisher Scientific Cat# A32814TR, RRID: AB_2866497) and donkey anti-mouse IgG Alexa Fluor 555 1:500 (Thermo Fisher Scientific Cat# A-31570, RRID: AB_2536180). Nuclei were labeled with DAPI mounting medium (P36962, Thermo Fisher Scientific, Waltham, MA, USA). Stained cells were analyzed on a Leica TCS SPE confocal microscope (Leica Microsystems, Wetzlar, Germany).

### 2.5. Cell Sorting

ic-GSCs were dissociated using Accutase (#A1110501, Thermo Fisher Scientific) into a single-cell suspension and labeled using double staining with anti-TRA-1-81 APC (#17-8883-42, Thermo Fisher Scientific) and SSEA4 Alexa Fluor 488 (#14-8843-80, Thermo Fisher Scientific). Cells were physically sorted using the FACSAria Fusion (BD Biosciences, Franklin Lakes, NJ, USA).

### 2.6. RNA-Sequencing Profiling and Analysis

Total RNA was extracted from GBM-DCs and TRA-1-81+/SSEA4+ ic-GSCs using the EZNA Total RNA Kit (R6834-01 Omega Bio-Tek, Norcross, GA, USA). Quality control, mRNA amplification, library preparation, and sequencing were performed at the CRG Genomics Unit (Centre for Genomic Regulation, Spain). Libraries were sequenced 2 × 50 + 8 + 16 bp on a HiSeq2500 sequencer (Illumina, San Diego, CA, USA). RNA-seq data were processed according to Doyle et al. protocol [19,20,21] using the R/DESeq2 package (v.1.28.1) [22]. RNA-seq counts were transformed to TPM using the R/countToFPKM package (v.1.2.0). GBM-DC and ic-GSC populations were represented as the mean values of all three GBM-DC replicates and ic-GSC lines, respectively. A representative centroid for the c-GSC population was generated by computing the mean of all genes in c-GSCs. A t-distributed stochastic neighbor embedding (t-SNE) of GBM-DC lines, ic-GSC lines, and a centroid representative of the c-GSC population including all c-GSCs DEGs was plotted using R/Rtsne (v.0.15).

### 2.7. DNA Methylation Profiling and Analysis

Genomic DNA was extracted from GBM-DCs and TRA-1-81+/SSEA4+ ic-GSCs using the Quick-DNA Microprep Kit (D-3020, Zymo Research, Irvine, CA, USA). Bisulfite conversion, amplification, and hybridization on the Infinium MethylationEPIC array BeadChip (850K, Illumina) were performed by the Epigenomic Services from Diagenode (Cat nr. G02090000). Raw DNAm data were processed and normalized to beta-values using the R/ChAMP package (v.2.18.3). All probes containing repetitive elements or single nucleotide polymorphisms (SNPs) were removed from downstream analyses. DNAm data were processed and normalized to beta-values using the R/ChAMP package (v.2.18.3). All probes containing repetitive elements or SNPs were removed from downstream analyses.

The promoter location (EPDnew Promoters) [23] was downloaded from UCSC Genome Browser Table [24,25]. All probes located at ±1.5 kb of a transcription start site were used to quantify the DNAm levels in gene promoters. OSN ChIP-seq peaks were acquired from the NCBI-GEO repository (GSE61475) [26]. OSN binding sites (OSNbs) were obtained using bedtools Intersect Intervals (Galaxy v.2.30.0.0). DNAm levels at ±10 kb from OSNbs were represented using deepTools (Galaxy v.2.30.0.0). The chromatin states were analyzed using the Epilogos visualization model (https://epilogos.altius.org/, accessed on 1 October 2021) based on epigenomic data sets across 833 biospecimens [27].

### 2.8. Statistics

For sc-RNA-seq data, the Wilcoxon test was used to analyze the differences in gene expression between c-GSC and tumor bulk cells. All *p*-values were corrected using the False Discovery Rate (FDR) method. Genes with corrected *p*-value < 0.05 and an absolute fold change > 0.5 were considered DEGs. For RNA-seq data, genes with an FDR corrected *p*-value below 10^−10^ and absolute log2 median of ratios >1.5 were considered DEGs. DNAm levels were standardized by dividing the value of each probe by the mean beta-value in each sample. All probes with differences in beta-value >1 and Wilcoxon-based *p*-value ≤0.1 were considered differentially methylated sites (DMS). The correlation matrices between GBM-DCs and ic-GSCs were plotted using the R/corrplot package (v.0.90). For both RNA-seq and DNAm data, representative GBM-DC and ic-GSC populations were determined as the mean value of all three GBM-DC replicates (GBM-DC1, GBM-DC2, GBM-DC3) and ic-GSC lines (ic-GSC#1, ic-GSC#3, ic-GSC#7), respectively.

## 3. Results

### 3.1. Identification and Characterization of Core-Glioma Stem Cells

We analyzed publicly-available scRNA-seq data from GBM patients (*n* = 28) [14], searching for undifferentiated cells harboring ESC-like signatures within the tumor mass. By pooling together all GBM patient cells (7930 cells), our analyses revealed the presence of 339 cells (4.27%) with the concurrent expression of the core ESC pluripotency markers OSN (Figure 1A, Link: https://coregliomastemcells.github.io/Fig1a.html, accessed on 13 December 2021). This population, described hereinafter as core GSC (c-GSC), was found in 26 out of 28 patients (92.8%), with an average presence of 4.22% ± 0.59 (Figure 1B).

Gene Ontology (GO) analysis revealed a significant downregulation of immune response in c-GSCs, mainly associated with antigen presentation pathways and the upregulation of stemness and lineage specification pathways (Figure 1C). In concordance with this observation, by measuring HLA-A, -B, and -C (HLA-ABC) expression in different tumor bulk cell populations ranging from triple-negative (OSN-) to triple-positive (OSN+), we observed a gradual downregulation of HLA-ABC genes towards the emergence of the c-GSC (OSN+) phenotype (Figure 1D). Downregulation of HLA type I and II genes on c-GSCs was also observed when analyzing each patient individually (Appendix A).

### 3.2. Reprogramming of Glioblastoma-Derived Cells Mimics the Core-Glioma Stem Cell Phenotype In Vitro

To understand the link between ESC features and immune evasion in GSCs, we reprogrammed GBM-DCs into induced c-GSCs (ic-GSCs) by the ectopic expression of the four pluripotency-related transcription factors OCT4, KLF4, c-MYC, and SOX2 (Yamanaka factors [15]; Figure 2A). Three independent ic-GSC lines (ic-GSC#1, ic-GSC#3, and ic-GSC#7) were generated from single-cell clones following serial dilutions and expanded into stable cell lines in the absence of doxycycline. To identify c-GSC molecular signatures in ic-GSCs, we analyzed the expression levels of the OSN factors and HLA-ABC by immunostaining (Figure 2B). Parental GBM-DCs were positive for SOX2 and HLA-ABC, whereas OCT4 and NANOG remained absent. On the other hand, in concordance with the tumor c-GSCs, all ic-GSC lines showed the expression of the three pluripotency-associated markers OSN and a strong downregulation of HLA-ABC.

### 3.3. Gene Expression Analysis of Induced Core-Glioma Stem Cells

To validate the reproducibility of ic-GSC lines (ic-GSC#1, ic-GSC#3, ic-GSC#7) and GBM-DC replicates (GBM-DC1, GBM-DC2, GBM-DC3), we generated gene expression profiles using an RNA-seq line. Showing a significant and consistent transcriptional reprogramming, the correlation analysis showed that each sample type clustered into two distinct groups representing ic-GSC and GBM-DCs (Figure 3A). Consistent with our observations by immunostaining, OCT4 and NANOG showed a significant upregulation and HLA-ABC significant downregulation in ic-GSCs (FDR corrected *p* <0.001) compared to the parental GBM-DCs (Appendix A). To evaluate whether ic-GSC resembles the transcriptional program of tumor c-GSC, we performed a t-SNE representation of the three ic-GSC lines, the three GBM-DC replicates, and a centroid representative of the tumoral c-GSC population. Our analysis showed clear segregation of all ic-GSC lines from the parental GBM-DC lines and an approximation to the c-GSC population (Figure 3B). This suggests that reprogramming GBM-DCs into ic-GSCs resulted in a distinct cellular state that resembles the c-GSC gene expression program. In line with this observation, ic-GSCs showed a strong agreement between differentially active gene pathways with c-GSCs, including significant upregulation of stemness and downregulation of immune-associated pathways (OR: 358.20, *κ* = 0.71, *p* < 0.0001; Figure 3C). This phenotype was also translated into the upregulation of additional stem-related genes (i.e., KDM1A and PROM1; Appendix A) and downregulation of immune-related genes (i.e., B2M, BCL2, IL6, NFKB1, among others; Appendix A).

### 3.4. Epigenetic Changes behind the ic-GSC Transcriptome Reprogramming

Similar to the transcriptomic analysis, the genome-wide DNAm profiling showed a significant shift between ic-GSC lines and GBM-DC replicates (Figure 4A). As expected, we observed that gene regulatory elements binding OSN showed significantly reduced DNAm levels, indicating an active involvement of these elements in ic-GSCs reprogramming (Figure 4B). Importantly, epigenetic activation involves gene regulatory elements that bind all three OSN factors, as individual sites or combinations of two of these factors did not show significant DNAm variations (Appendix A).

Integration of RNA-seq and DNAm data revealed that downregulation of the HLA-C gene correlated with increased DNAm levels at its promoter region (Figure 4C). In contrast, DNAm at HLA-A and HLA-B promoter regions remained unchanged. On the other hand, endogenous SOX2 and NANOG genes presented no differential DNAm at their promoters, whereas the endogenous OCT4 promoter region showed reduced DNAm levels. To identify potential cis and trans gene regulatory elements whose activation may impact HLA-ABC regulation in c-GSCs, we evaluated the DNAm levels on a 1.7 Mb genomic region (chr6: 29,685,442–31,426,907), encoding a cluster of HLA class I genes (from HLA-A to HLA-F; Figure 4D). To contextualize the potential functional impact of differentially methylated genomic regions, we analyzed the chromatin states of the entire region based on a public compendium of epigenomic maps across 833 specimens [27]. We observed that regions with significant hypermethylation in ic-GSCs overlapped chromatin segments with variable states (Figure 4D red rectangles). Depending on the cell type and state, these regions can act as enhancer elements (light green and orange pixels), be repressed (grey pixels), or remain quiescent (white pixels). We compared chromatin states between stem and non-stem samples to evaluate associations with the ic-GSC phenotype. This analysis showed a decrease in active transcription start sites (TSS; red pixels) marks in HLA-ABC genes, along with an increment of active TSS at the OCT4 (POU5F1) gene (Figure 4D black rectangles, Appendix A). More importantly, hypermethylated regions in ic-GSCs showed a clear association with stem-related polycomb repressed and bivalent poised marks at the HLA-B and HLA-C promoter regions and nearby regulatory elements (Appendix A).

## 4. Discussion

The idea that tumors originate from embryonic-like cells is a longstanding hypothesis originally conceived in the 19th century after observing histologic similarities between cancer and embryonic tissue under the microscope [28]. Over the years, mounting evidence has led researchers to theorize that tumors may arise from tissue-specific stem cells that undergo malignant transformations [29]. In GBM, the isolation and characterization of GSCs have been carried out by culturing primary tumor cells under neural stem cells in serum-free media conditions [30]. Nevertheless, purification, expansion, and characterization of GSCs remain a challenging endeavor. In addition, there have been inconsistencies concerning the GSC model due to the lack of consensus on culture methods and specific GSC markers used among different research groups [31]. Based on the original concept of the embryonal origin of cancer, and thanks to the advent of scRNA-seq technology, we were able to identify a population of GBM cells with concurrent high levels of the core ESC pluripotency factors OCT4, SOX2, and NANOG.

One way tumors escape from immune recognition is through the downregulation of HLA molecules [32]. Several reports have identified a defect in HLA class I in gliomas associated with immune evasion mechanisms [33,34]. We identified a significant downregulation of HLA-ABC genes in c-GSCs compared with the rest of the tumor cells. Unfortunately, due to the lack of complete profiling in the scRNA-seq specimens, our study was limited to identifying associations between the c-GSC content and relevant GBM markers, such as IDH1/2 status and MGMT promote methylation, among others. These findings led us to speculate that this c-GSC population could have a critical role in GBM tumor growth and invasion, tumor immune evasion, and ultimately relapse after treatment. It would be important to investigate if the unsuccessful IT is associated with a higher percentage of c-GSCs in the tumor mass. Despite the sub-optimal results obtained from clinical trials, multiple efforts are being made to develop novel strategies for GBM IT, including neoantigen vaccines, oncolytic viruses, CAR-T cells, and immune-checkpoint inhibitors [35,36]. Thus, targeting c-GSCs might help elucidate GBM immune evasion-related molecular mechanisms and may provide potential alternatives to improve IT outcomes.

A challenge of the current study was the difficulty to isolate and expand c-GSCs in vitro, mainly due to sample accessibility and lack of standardized protocols. While these cells represent 4.22% ± 0.59 and are present in 92.8% of the GBM patients, prior GSC studies may have missed the existence of these embryonic-like cells due to the culturing conditions. To overcome this, we used iPSC technology to reconstitute a c-GSC model in cell culture (Figure 2A). The generation of iPSCs aims to reprogram somatic cells into a pluripotent state, similar to ESCs from the early embryo [15]. Our goal was to reprogram GBM cells and generate cells with a c-GSC-like phenotype that we named ic-GSCs. Similar to the gene expression profile of c-GSCs, we confirmed that ic-GSCs expressed OCT4, SOX2, and NANOG but lacked HLA-ABC expression (Figure 2B). Pluripotent ESCs and iPSCs are immune-privileged by presenting the low expression of HLA class I genes [37,38], which is associated with an immunological tolerance mechanism of the early embryo to avoid immune rejection by the host [39]. A similar mechanism could occur during tumorigenesis, where the tumor may take advantage of immune-privileged undifferentiated cells to escape from immune system recognition.

To further validate our ic-GSC model, we compared transcriptomic-level profiles of ic-GSC and c-GSC populations (Figure 3C). Our analysis showed that reprogramming GBM-DCs into ic-GSCs resulted in a distinct cellular state that recapitulates the gene expression program of c-GSCs from tumor bulk. In fact, we observed that ic-GSCs have a transcriptome profile more similar to c-GSCs than to its parental GBM-DCs and specifically showed a strong agreement between differentially active gene pathways, both sharing an upregulation of stemness and a downregulation of immune-associated pathways. In addition to changes in gene expression, we investigated epigenetic marks potentially involved with the c-GSC aggressive phenotype. We generated DNAm maps from GBM-DCs and ic-GSCs to identify epigenetic changes involved in ic-GSC reprogramming. These maps suggest that the shift in the phenotype is orchestrated by OSN target genes, as our DNAm analysis revealed a significant hypomethylation at OSN binding sites. Moreover, while the HLA-C promoter region showed a modest DNAm increment in ic-GSCs, the HLA-A and HLA-B promoter regions remained unchanged. These findings indicate that DNAm alone is not the main mechanism for HLA-ABC downregulation, and other epigenetic mechanisms might be involved.

In addition, our chromatin state analysis in stem-related specimens confirmed a slight DNAm variation at the HLA-ABC promoter regions and a significant hypomethylation of the OCT4 (POU5F1) gene (Figure 4D black rectangles). Moreover, our analysis also revealed the presence of polycomb repression and bivalent marks on the HLA-B and HLA-C promoter regions. A similar phenomenon has recently been observed by Chaligne et al. [40], where polycomb activity may establish bivalent domains at hypomethylated promoters in GBM stem-like cells. Accordingly, Burr et al. [41] reported that polycomb maintains transcriptional repression of HLA class I genes, keeping them poised by bivalent histone modifications in ESC and cancer cell lines. Thus, the embryonic traits of c-GSCs may lead to an epigenetic reprogramming that results in the silencing of HLA genes under a poised state to achieve an immune-evasive phenotype.

## 5. Conclusions

Our findings support the existence of a subset of stem cells in GBM with embryonic-like features and an attenuated immune phenotype. Here, we provide an in vitro model that mimics the c-GSC phenotype that provides a valuable resource to study the mechanisms governing c-GSCs immune evasion and may set the basis to improve the current GBM patient’s response to IT.

## Figures and Tables

**Figure 1 cancers-14-02070-f001:**
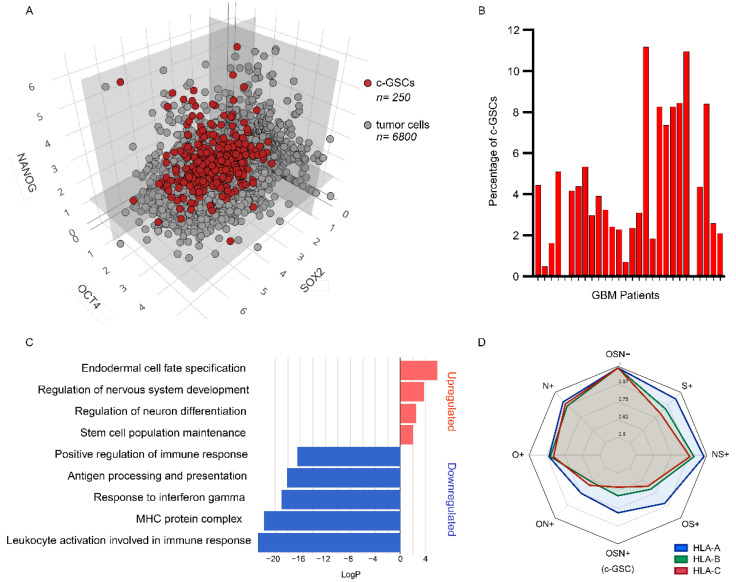
Core glioma stem cell population in glioblastoma samples. (**A**) Three-dimensional scatter plot of GBM single-cells (*n* = 7930) from 28 patients based on the expression levels of OCT4 (*x*-axis), NANOG (*y*-axis), and SOX2 (*z*-axis). Red dots represent c-GSCs co-expressing OCT4, SOX2, and NANOG (*n* = 339). Gray dots represent the remaining cells from tumor bulk (*n* = 7591). An interactive version of this data can be found: https://coregliomastemcells.github.io/Fig1a.html, accessed on 13 December 2021. (**B**) Bar plot showing the percentage of c-GSCs in each GBM patient. (**C**) GO analysis of biological pathways upregulated (red) and downregulated (blue) in c-GSCs. (**D**) Radar plot representing the expression levels of HLA-A, -B, and -C genes on different GBM single-cell populations based on the expression of OCT4, SOX2, and NANOG (OSN).

**Figure 2 cancers-14-02070-f002:**
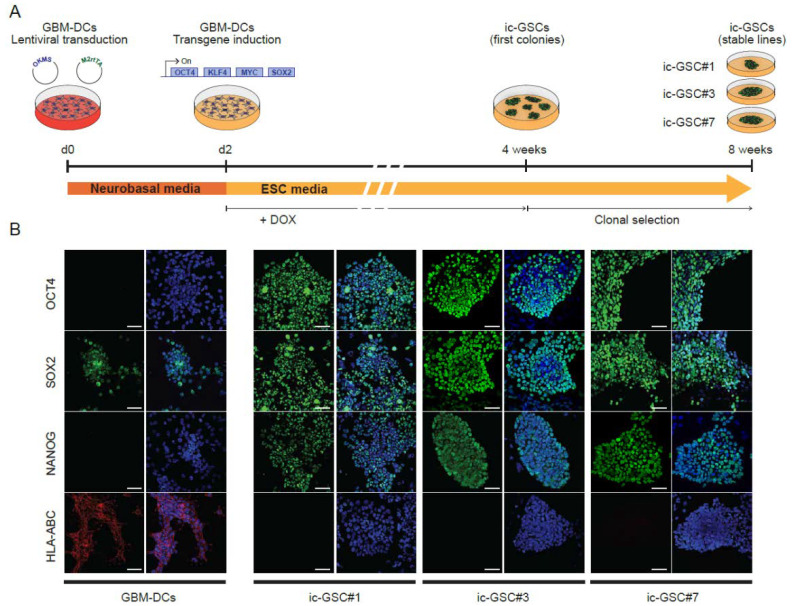
Characterization of an in vitro induced core glioma stem cell model generated from glioblastoma patient-derived cells. (**A**) Workflow depicting the reprogramming of GBM-DCs into ic-GSC stable lines. (**B**) Immunostaining of OCT4, SOX2, NANOG, and HLA-ABC in GBM-DCs and three independent ic-GSC clones: ic-GSC#1, ic-GSC#3, and ic-GSC#7. Scale bars correspond to 50 µm.

**Figure 3 cancers-14-02070-f003:**
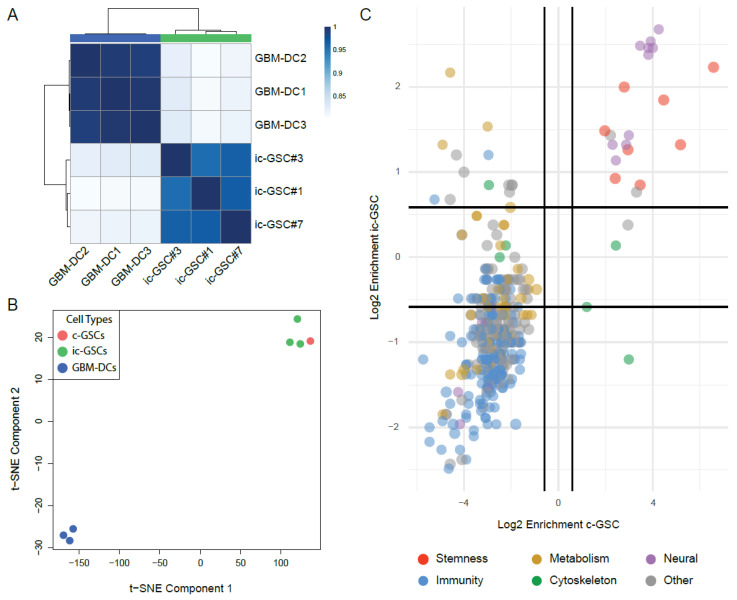
Transcriptomic analysis of induced core glioma stem cells. (**A**) Correlation matrix for three GBM-DC replicates and three ic-GSC lines based on gene expression profiles. (**B**) t-SNE plot of three GBM-DC replicates (blue), three ic-GSC lines (green), and a representative centroid of the c-GSC population (red) using gene expression profiles (**C**) Scatter plot of enriched pathways in representative ic-GSC and c-GSC populations and colored based on their biological network classification. GBM-DC replicates: GBM-DC1, GBM-DC2, and GBM-DC3; ic-GSC lines: ic-GSC#1, ic-GSC#3, and ic-GSC#7.

**Figure 4 cancers-14-02070-f004:**
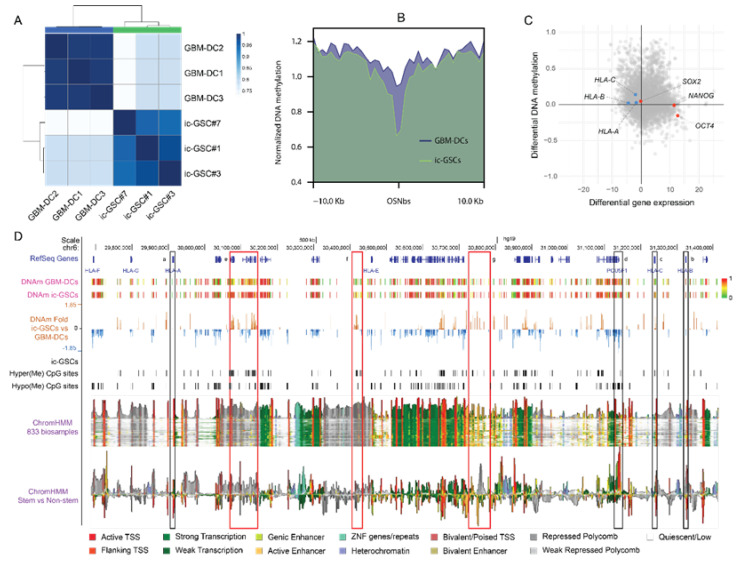
Epigenomic analysis of induced core glioma stem cells. (**A**) Correlation matrix for three GBM-DC replicates and three ic-GSC lines based on DNAm profile. (**B**) Profile plot representing the DNAm levels of GBM-DCs (blue) and ic-GSCs (green) around (±10 kb) consensus OCT4, SOX2, and NANOG binding sites (OSNbs). (**C**) Starburst plot including DMS (*y*-axis) and DEG (*x*-axis) in representative ic-GSC and c-GSC populations, highlighting stem-related factors OCT4, SOX2, and NANOG (red) and immune-associated antigen presentation genes HLA-A, -B, and -C (blue) (**D**) Representation of a 1.7 Mb genomic region (chr6: 29,685,442–31,426,907) encoding HLA-ABC genes and genes relevant for ic-GSC reprogramming. The view includes RefSeq Genes from NCBI, DNAm levels in GBM-DCs and ic-GSCs, DNAm fold change in ic-GSCs vs. GBM-DCs, hyper- and hypomethylated CpG sites in ic-GSCs, chromatin state (ChromHMM) in 833 biosamples, and chromatin state in stem versus non-stem biospecimens. HLA-ABC and OCT4 (POU5F1) promoter regions are represented as black rectangles. Hypermethylated regions in ic-GSCs are represented as red rectangles.

## Data Availability

RNA-seq data that support the findings in this study are openly available in the ArrayExpress database at EMBL-EBI (www.ebi.ac.uk/arrayexpress, accessed on 9 December 2021) under accession number [E-MTAB-10977] and [E-MTAB-10978].

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
