# Peer review of "Glioblastoma Embryonic-like Stem Cells Exhibit Immune-Evasive Phenotype"

_cancers, 2022, doi:10.3390/cancers14092070_

Round 1

Reviewer 1 Report

These transcriptomic and epigenetic studies characterized inducible glioblastoma stem cells by correlating the expressions of immune-related genes such as HLA-ABC with OCT4, SOX2 and NANOG. In this article, the authors discovered that the expressions of three stem cell markers effectively depressed HLA-associated genes resembling the patterns of the core GBM cells from patients. The studies were well designed, and the demonstration of the results was satisfactory, also supported by the supplements.  

Author Response

Thank you very much for your feedback.

Reviewer 2 Report

Dear All,

Glioblastomas (GBMs) normally relapse due to their subpopulation of glioma stem cells (GSCs) within the tumor.  The authors carried out experiments to analyze the stemness hierarchy of GSCs.  Their results from RNAseq revealed that GSCs, named core GSCs, have a similar profile to embryonic stem cells.  In addition, antigen-presenting proteins (HLAs) were down-regulated in GSCs.  These findings were corroborated induced by core GSCs (ic-GSCs) derived from GBMs. The HLAs were repressed in GSCs by DNA methylation with polycomb repression features. 

The manuscript is acceptable when the minor revision is corrected.

1)  Supplementary Figure 3g is missing.

Author Response

Thank you very much for your feedback. Supplementary Figure 3g (CTCF binding sites) has been removed from Supplementary Materials. We removed Figure S3g from the figure caption in our original submission but forgot to remove it from the Supplementary Materials main text.  We decided to discard it because there were no significant changes and wasn’t relevant enough (attached).  As you can see in this plot, the blue line represents GBM-DCs and the green line represents the ic-GSCs. If you consider this information relevant for a better understanding of the study, we would be happy to include it in an additional revision of the manuscript.

Reviewer 3 Report

Authors found that  a 33 GSC population (4.22% ±0.59) exhibiting concurrent high expression of ESC markers and downregulation of immune-associated pathways, named c-GSCs. In vitro ic-GSCs presented high expression of ESC markers and downregulation of antigen presentation HLA proteins.  This paper is interesting, however following points should be taken into consideration.

  1. The profiling of GSCs, including  IDH-status, H3K27M, MGMT and so on, is not clear.
  2. Authors should show direct evidence which HLA proteins work as GSCs immune-evasive mechanisms

Author Response

Thank you very much for your feedback.

1- The profiling of GSCs, including IDH-status, H3K27M, MGMT, and so on, is not clear.

We observed no differences in IDH gene expression between c-GSCs and the remaining tumor bulk cells from sc-RNAseq data, and unfortunately, we have not been able to infer IDH mutation statuses from the data provided in this study. The public GBM database we used contained sc-RNAseq data only, so we were not able to evaluate epigenetic marks such as H3K27M.

2- Authors should show direct evidence which HLA proteins work as GSCs immune-evasive mechanisms

It has been reported that loss of class I HLA function is an important mechanism for tumor immune evasion (we have included a reference in the manuscript [32]). However, there is no evidence of which particular HLA protein is responsible for tumor cells escape from immunotherapy. Our analysis revealed that all class I HLAs were downregulated in c-GSCs (q value < 10-5). Future experiments on candidate epigenetic factors targeting the HLA locus and HLA-specific knock-out experiments may shed some light on specific HLA proteins and immune-evasive responses in GSCs.